# Cleaning of Phototrophic Biofilms in a Show Cave: The Case of Tesoro Cave, Spain

**Valme Jurado** [1,*] **, Mariona Hernandez-Marine** [2] **, Miguel Angel Rogerio-Candelera** [1] **, Francisco Ruano** [3] **, Clara Aguilar** [3] **, Juan Aguilar** [3] **and Cesareo Saiz-Jimenez** [1]

1   Instituto de Recursos Naturales y Agrobiologia, IRNAS-CSIC, 41012 Sevilla, Spain; marogerio@irnase.csic.es (M.A.R.-C.); saiz@irnase.csic.es (C.S.-J.)
2   Facultad de Farmacia, Universidad de Barcelona, 08028 Barcelona, Spain; marionahernandez@ub.edu
3   Agora S.L., 41928 Sevilla, Spain; frgdepau@hotmail.com (F.R.); aguilarlinaresclara@yahoo.es (C.A.); agora2@telefonica.net (J.A.)
*   Correspondence: v.jurado@csic.es

**Abstract:** Show caves have different grades of colonization by phototrophic biofilms. They may receive a varied number of visits, from a few thousand to hundreds of thousands of visitors annually. Among them, Tesoro Cave, Rincon de la Victoria, Spain, showed severe anthropic alterations, including artificial lighting. The most noticeable effect of the lighting was the growth of a dense phototrophic community of cyanobacteria, algae and bryophytes on the speleothems, walls and ground. The biofilms were dominated by the cyanobacterium *Phormidium* sp., the chlorophyte *Myrmecia israelensis*, and the rhodophyte *Cyanidium* sp. In many cases, the biofilms also showed an abundance of the bryophyte *Eucladium verticillatum*. Other cyanobacteria observed in different biofilms along the cave were: *Chroococcidiopsis* sp., *Synechocystis* sp. and *Nostoc* cf. *edaphicum*, the green microalgae *Pseudococcomyxa simplex*, *Chlorella* sp. and the diatom *Diadesmis contenta*. Preliminary cleaning tests on selected areas showed the effectiveness of hydrogen peroxide and sodium hypochlorite. A physicochemical treatment involving the mechanical removal of the thickest layers of biofilms was followed by chemical treatments. In total, 94% of the surface was cleaned with hydrogen peroxide, with a subsequent treatment with sodium hypochlorite in only 1% of cases. The remaining 5% was cleaned with sodium hypochlorite in areas where the biofilms were entrapped into a calcite layer and in sandy surfaces with little physical compaction. The green biofilms from the entire cave were successfully cleaned.

**Keywords:** cyanobacteria; chlorophytes; bryophytes; surface cleaning; hydrogen peroxide; sodium hypochlorite

## 1. Introduction

Caves are heterogeneous karst systems composed of different compartments. As one progresses from the outside to inside the cave, it can be observed that the microbial populations colonizing the walls and ceilings are different, and separate into different zones. In consequence, there is a transition between adjacent ecological systems that involves a change in the structure and composition of the community and is limited by their spatial distribution. Phototrophic communities (cyanobacteria, algae, lichens, bryophytes and lower plants) are usually abundant in the area closest to the outside, generally illuminated by natural light, and disappear towards the deepest area, depending on the decrease in the light gradient.

Cyanobacteria and algae are especially successful microorganisms in subterranean environments due to the high humidity and carbon dioxide concentration and the very low light intensity in which they thrive. Some authors call these biofilms lampenflora [1–3]. The lighting in show caves and other underground environments generates a series of problems, derived from the growth of phototrophic communities, which generally represents a serious

ecological distortion and an aesthetic problem due to the presence of biofilms with green to black colorations colonizing the speleothems and walls.

The problem of the appearance of green biofilms in caves with Paleolithic paintings is not new. A few cases have been documented in the scientific literature and the phototrophic communities threaten the Paleolithic paintings.

Lascaux Cave in France had to be closed at the beginning of the 1960s due to the invasion of the alga *Bracteacoccus minor* as a result of the intense flow of visitors and hours of artificial light. The biofilms were removed with antibiotics and formaldehyde [4,5]. Altamira Cave was closed in 2002 due to the occurrence of cyanobacteria and algae on the paintings [6,7]. A third case was recently reported in El Castillo Cave, with the occurrence of biofilms in the Polychromes Panel [8]. In this cave, chlorophytes, specifically *Jenufa aeroterrestrica*, a new species described in 2015 [9], as well as *Coccomyxa* sp. were found in the biofilms, but no cyanobacteria. In Altamira and El Castillo caves, with initial stages of phototrophic colonization, lighting was turned off.

Other caves showed extensive green biofilms on speleothems, but far from the paintings, such as in Tito Bustillo Cave, Asturias, whose stalactites and stalagmites were abundantly colonized by phototrophic microorganisms and, in particular, the sediments showed an exuberant colonization by the calcifying cyanobacterium *Scytonema julianum*, common in caves, tombs and catacombs [10–12].

Del Rosal et al. [13,14] found *Jenufa* sp. in Nerja Cave, with abundant phototrophic biofilms on the speleothems and walls. In areas with water percolation (occasional or habitual), chlorophytes (*Jenufa* sp. and *Desmococcus endolithicus*) were commonly identified, while cyanobacteria and the red alga *Cyanidium* sp. abounded in drier areas.

The Andalusian caves, Gruta de las Maravillas, Murcielagos, Tesoro and Nerja, to mention a few caves, exhibit different grades of phototrophic biofilm development. All of them can be accessed and receive a varied number of visits, from a few thousand to hundreds of thousands of visitors annually. Among them, Tesoro and Nerja caves stand out due to the abundance of phototrophic biofilms. While Nerja Cave biofilms have been thoroughly studied [15], the composition of the biofilms from Tesoro Cave is relatively unknown.

Tesoro Cave (Figure 1) is situated in Rincon de la Victoria, about 11 km from Malaga, Spain. The cave is located in a calcareous massif (Cantal Alto), formed on a thick bed of about 70 m of limestone. The cave is about 80 m above sea level. The Cantal hill is widely karstified with an abundance of small cavities and two main caves, Tesoro and Rincon de la Victoria, the first located in the central area of Cantal Alto massif, and the second more to the south.

The climate is semiarid and mesothermal, with small surplus of water in winter. The water that reaches the karst is scarce; a significant quantity is lost by evapotranspiration, and the remainder is lost in the urbanized area surrounding the cave entrance. The only water supply to Cantal karst is reduced to direct rain infiltration with a few dripping points in the cave; the response time to rain has 2–3 h of delay. This explains why the generation of speleothems is so scarce in the cave [16].

The cave measurements of $CO_2$, temperature and relative humidity showed particular features. In Sala del Lago and Sala del Volcan, the $CO_2$ concentrations were from 200 to 550 ppm, the temperature 14.8–16.8 °C, and the relative humidity 80% to 90%, all of which are normal values for a karstic cavity [5–8,13–15]. However, the non-visited area (Galeria de Breuil) shows anomalous concentrations of $CO_2$ [16], such as 20,000 ppm, the temperature rises to 22.7 °C, and the relative humidity reaches 100% saturation. These values are characteristic of environments with poor ventilation; in other words, the karstic system seems to behave, in this Galeria, as a confined and almost impermeable enclosure that presents a small renewal of air.

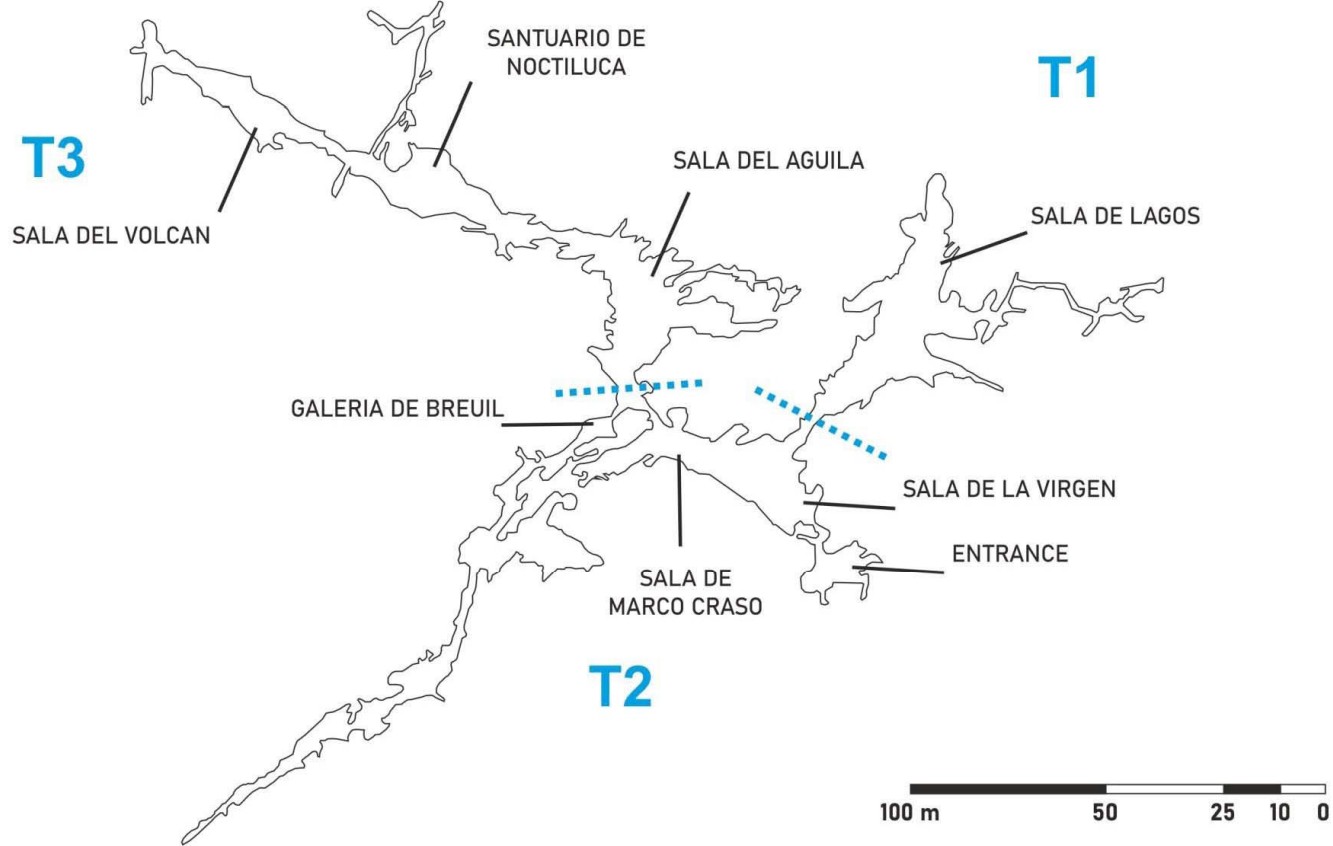

**Figure 1.** Map of Tesoro Cave. T1–T3 represent the three sectors into which the cave was divided for study purposes.

Tesoro Cave has experienced severe anthropic alterations. On the surface, the area was urbanized with roads, parking plots and buildings. Inside the cave, strong alterations for facilitating visits are observed, including the removal of sediments, building materials extraneous to the karst, artificial lighting, the construction of an artificial lake, an elevator, and a wide staircase [16]. Surface corrosion was favored by the microenvironmental conditions in some galleries. The most noticeable effect of lighting was the growth of a dense phototrophic community of cyanobacteria, green algae and bryophytes on the speleothems, walls and ground, near the lighting points and wet areas. The remains of wood and an old electric installation are also evident.

For study purposes, the cave was divided into three sectors: T1, mainly formed by a wide hall, Sala de Lagos, with an artificial lake; T2, centered by the main entrance to the cave with an elevator and staircase as well as halls and galleries at both sides to give access to zones T1 and T3; and T3, ending in a deep hall, Sala del Volcan (Figure 1). Here, we report a study on the phototrophic community in Tesoro Cave, the cleaning of biofilms and the used protocols.

## 2. Materials and Methods

Phototrophic biofilm samples were scraped off with a sterile scalpel, placed into 1.5 mL sterile Eppendorf tubes and stored at 4 °C until arrival at the laboratory.

Taxonomic identifications were based on the morphological characters of the whole biofilm and on colonies grown on 15% agar plates with medium BG11 (Sigma-Aldrich, Steinheim, Germany) or Bold's Basal Medium (BBM) (Sigma-Aldrich, Steinheim, Germany) and incubated at room temperature and a low light intensity (semi-closed window oriented to the north) [17,18]. The identification of the different microalgae and cyanobacteria was based on the available taxonomic keys and confirmed in AlgaeBase [19–24]. Molecular biology techniques were applied

for the identification of *Cyanidium* sp. DNA was extracted using the FastDNASPIN for Soil Kit (MP Biomedicals, Illkirch, France). The amplification of 16S gene sequences was performed using the cyanobacteria-specific primer pair, Cya106F (5′-CGGACGGGTGAGTAACGCGTGA-3′) and Cya 781R (5′-GACTACTGGGGTATCTAATCCCWTT-3′) [25]. The PCR amplification protocol consisted of the following thermal conditions: 95 °C for 2 min; 35 cycles of 95 °C for 15 s, 60 °C for 15 s, 72 °C for 2 min; and a final step of 72 °C for 10 min. DNA libraries of PCR-amplified products were constructed using the TOPO-TA cloning kit (Invitrogen, Carlsbad, CA, USA) according to the manufacturer's recommendations. Plasmids were extracted with the JetQuick Plasmid Miniprep Spin kit (Genomed, Löhne, Germany), following the manufacturer's protocol, and sequenced by Secugen Sequencing Services (CSIC, Madrid, Spain). The identification of phylogenetic neighbors was determined using the BLASTN algorithm [26]. The 16S rRNA gene sequences of clones and its closest related sequences were multiplied and aligned using CLUSTAL W [27]. The phylogenetic tree was constructed using the maximum-likelihood method [28] and Kimura's two-parameter model with a discrete Gamma distribution in MEGA version 11 [29]. A bootstrap analysis of 1000 re-samplings was used to evaluate the robustness of the tree. All sequences were submitted to GenBank with consecutive accession numbers ON619559–ON619566.

The direct observation of phototrophic biofilms and isolate colonies from enrichment cultures was carried out using an Axioplan microscope (Carl Zeiss, Oberkochen, Germany) and the images captured with an AxioCam MRc5 digital camera and processed with Axiolan LE software and a scanning electron microscope (Quanta 200 FEI Φ EDAX) [13,14]. The samples were subjected to pre-fixation with acrolein vapor and post-fixation with osmium tetroxide vapor. The samples were double coated with carbon and sputtered gold.

Several treatments were tested for biofilm removal and cleaning. The treatments applied for cleaning the whole cave included mechanical removal by brush and cleaning with liquid nitrogen, with hydrogen peroxide and/or sodium hypochlorite.

Physicochemical cleaning was a two-phase treatment: the first physical and the second chemical. The first consisted of the mechanical removal of the thickest layers of biofilms using a spatula and a brush. This treatment was conditioned by the type of biofilm and the rock surface. In fact, this was applied to dense biofilms showing abundance of bryophytes and/or algae that could easily be removed from the rock surfaces. The application was never done dry, but rather by means of previous hydration of the surfaces. Once finished, a chemical treatment was applied, which was conditioned by the typology of the surface (solid rock surfaces), the type of biofilms (residual algae and/or bryophyte protonema not removed by brushing) and the environmental conditions (dry or wet surfaces). The products used were as follows.

Cleaning with hydrogen peroxide: hydrogen peroxide at 10% in water was very effective in dry areas. It was applied by impregnation with a brush or by spraying with nebulizers. In the case of more humid areas and with more irregular textures, it was applied in a percentage of 50% in aqueous solution. This was done by first performing various tests. We observed that in areas with high humidity (due to condensation or seepages), the product was diluted, lowering the concentration and losing effectiveness. This chemical was the most widely used.

Cleaning with sodium hypochlorite: cleaning with 10% sodium hypochlorite in aqueous solution was used in very porous or textured areas where cracks abounded and the conditions, in all ways, were very extreme, and in areas of very high biofilm development, especially in areas located around light sources. It was applied by impregnation with a brush.

Subsequent chemical cleaning: this treatment was mainly applied in areas where the biofilms were entrapped into a calcite layer and on sandy surfaces with little physical compaction. Two different scenarios were considered, namely, fragile and calcified zones.

Fragile zones: small stalactites, or very fragile surfaces. The treatment was applied with a nebulizer in areas where hydrogen peroxide was not effective enough.

Calcified zones: applied in areas where the biofilm was covered by calcite crystals, showing an intense green hue. To access the biofilms, pads impregnated with sodium hypochlorite were placed on the mineral layer, which reduced the evaporation of the product and facilitated its action.

## 3. Results and Discussion

### 3.1. Composition of Biofilms in Different Cave Sectors

#### 3.1.1. Sector T1

The cave was divided into three sectors with different ecological and environmental characteristics. Sector T1, largely represented by Sala del Lago, was influenced by the artificial lake occupying almost the entire sector, which provided high relative humidity, near saturation. In addition, the connection with the exterior was relevant through the water. This and the lighting produced abundant phototrophic biofilms composed of cyanobacteria, microalgae and bryophytes near the lighting lamps. The biofilms were dominated by the cyanobacterium *Phormidium* sp. (Figure 2B), the chlorophyte *Myrmecia israelensis* (Figure 2A) and the rhodophyte *Cyanidium* sp. It is remarkable that *Cyanidium* sp. was only found in this sector and not in the biofilms from the rest of the cave.

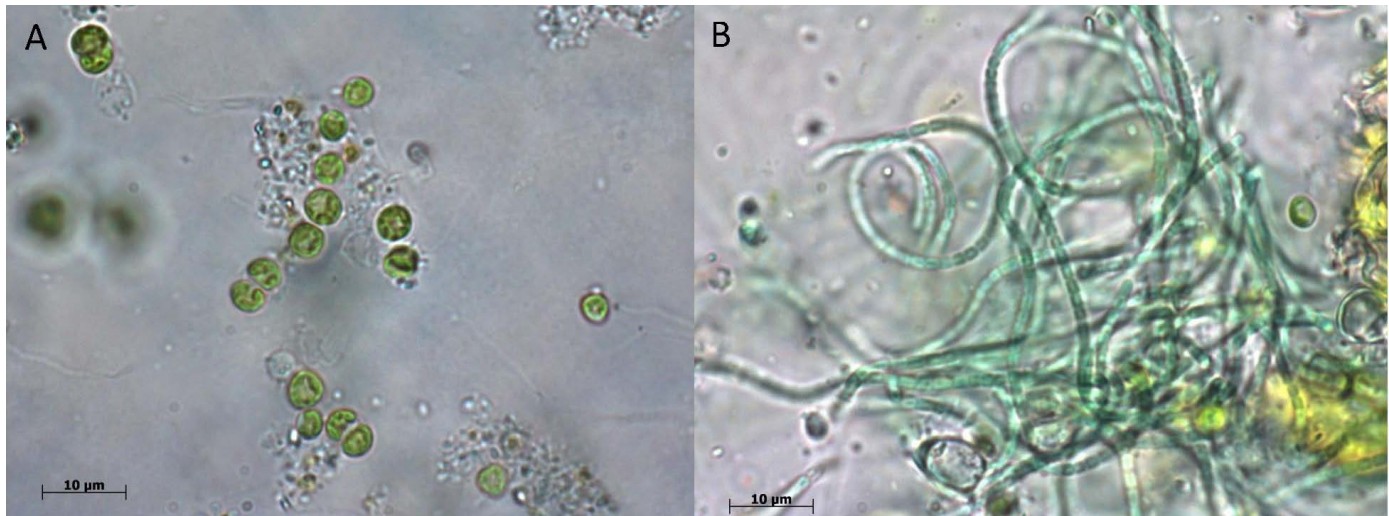

**Figure 2.** (**A**) Cells of *Myrmecia israelensis*. (**B**) Filaments of *Phormidium* sp. Scale bar: 10 μm.

The presence of *Cyanidium* sp. in Tesoro Cave, which was also observed in Nerja Cave some 50 km away [14], deserves a special mention. The reports on the genus *Cyanidium*, identified by microscopy and morphology and retrieved from non-extremophilic habitats, are very scarce, but not those from extreme and acidophilic environments, where *Cyanidium caldarium* is well represented [30,31]. *Cyanidium* sp. would not correspond to *C. caldarium* due to its very different habitat and ecology. In fact, *Cyanidium* strains from caves cluster into a separate monophyletic lineage within the class *Cyanidiophyceae* [32].

The first morphospecies of the genus *Cyanidium* that did not inhabit acidic ecosystems or volcanic areas were described from samples from caves (*Cyanidium chilense*) and from fissures in coastal rocks, both in Chile [33]. Subsequently, by means of molecular analysis, three species of non-extremophilic aerophytic *Cyanidium* have been described, two of them *Cyanidium* sp. Monte Rotaro and *Cyanidium* sp. Sybil Cave, inhabitants of Italian caves [34,35], and the third, *Cyanidium* sp. Atacama, in a cave on the Chilean coast [36].

The sequences of clones related to the genus *Cyanidium* identified in Tesoro Cave present identities of 95.09%, 94.75% and 92.76% with the strains from Atacama, Monte Rotaro and Sybil Cave, respectively (Figure 3). In the phylogenetic tree, the Tesoro Cave sequences form a robust clade far from previous *Cyanidium* isolates, suggesting that they could represent a different species of the genus.

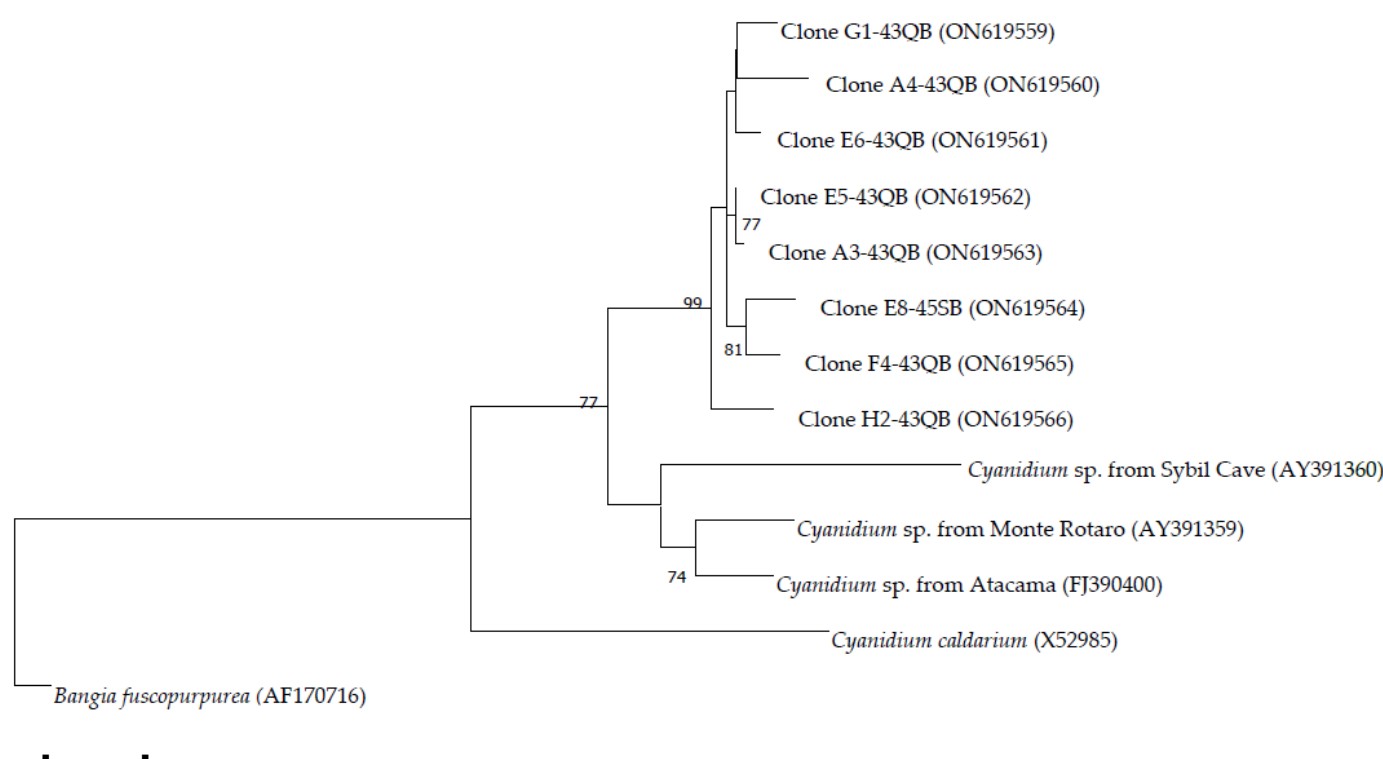

**Figure 3.** Maximum-likelihood phylogenetic tree based on 16S rRNA gene sequences showing the relationship of clones from phototrophic biofilms collected in Tesoro Cave to isolated species of the genus *Cyanidium*. Bootstrap values (>50%) are expressed as percentages of 1000 replicates. The 16S rRNA gene sequence of *Bangia fuscopurpurea* was used as the outgroup. Bar: 0.02 substitutions per nucleotide position.

In many cases, the biofilms also showed abundance of the bryophyte *Eucladium verticillatum*. The cause was the intense illumination by means of incandescent lamps, as shown in Figure 4. *Eucladium verticillatum* usually live outdoors in very pronounced shaded areas and produce rhizoid buds. However, this bryophyte is common in limestone caves, where it grows at varied ranges of light irradiation [37,38]. Whitehouse [39] reported the presence of *E. verticillatum* in Altamira Cave in 1955. We also find it in Encajero Cave, in Quesada, Jaen, with a thick spongy growth giving rise to considerable tufa formations, associated with *Stichococcus bacillaris* [40].

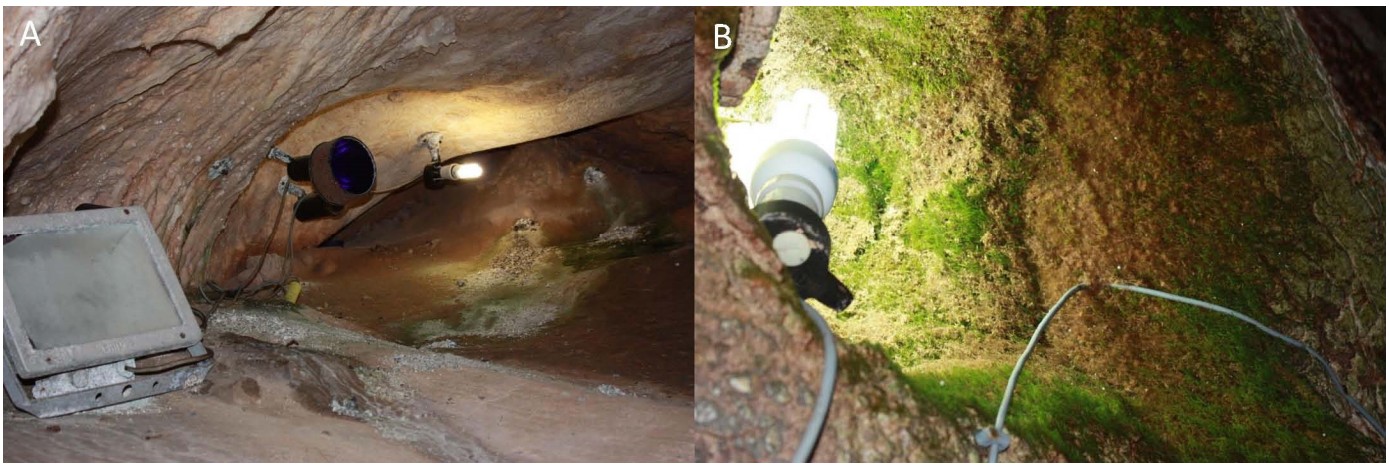

**Figure 4.** Growth of the bryophyte *Eucladium verticillatum*. (**A**) Sala del Volcan. (**B**) Sala del Aguila.

In the studied cases, cells of *M. israelensis*, *Phormidium* sp. and *E. verticillatum* formed a dense biofilm. Scanning electron microscopy images of the biofilms showed a dense network of *M. israelensis* cells (Figure 5A) and filaments of *Phormidium* sp. (Figure 5B). The cells of *M. israelensis* and *Phormidium* sp. are usually attached to each other by appendages. Some cells are lodged in depressions that suggest an active mechanism of calcite dissolution.

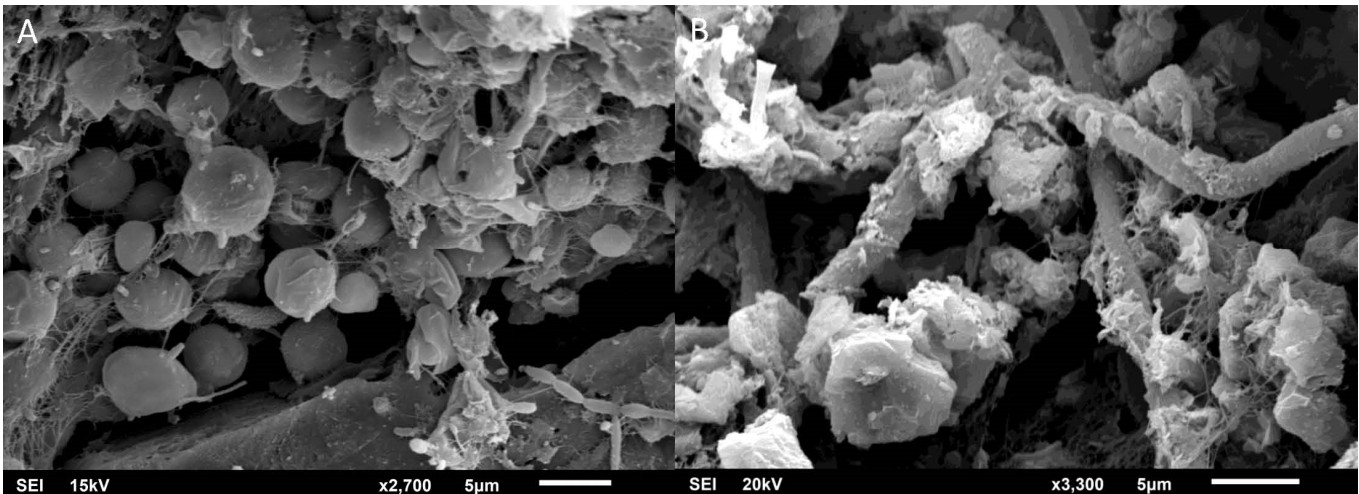

**Figure 5.** Scanning electron images of biofilms. (**A**) Cells of *Myrmecia israelensis*. (**B**) Filaments of *Phormidium* sp. Scale bar: 5 μm.

### 3.1.2. Sector T2

This sector is characterized by the location of an elevator and the staircase in the center, connecting with the entrance, and with halls and galleries at both sides connecting with sectors T1 and T3. Therefore, this sector was anthropically modified and showed high environmental changes with respect to the rest of the cave.

In Sala de Marco Craso, there was a phototrophic colonization occupying large extensions. The light microscope as well as the scanning electron microscope study revealed that the phototrophic community was almost exclusively formed by *M. israelensis*. The substratum was heavily corroded, and on it had been deposited masses of the algae.

In other biofilms from Sala de Marco Craso, several species of cyanobacteria were observed: *Synechocystis* sp. and *Nostoc* cf. *edaphicum* (Figure 6A,B) and the microalgae *Pseudococcomyxa simplex*, *Chlorella* sp. and sporocysts of *M. israelensis* (Figure 6C). Sporocysts are cells within which spores are formed. The identification of a species of *Nostoc* is interesting, since it indicated that there is nitrogen fixation from the air in the cave by cyanobacteria, which would enrich the niche and provide this element for the growth of non-nitrogen-fixing heterotrophic microorganisms. The violet color of Figure 6A is due to the pigment phycocyanin that produces *Nostoc* and that adheres to membranes.

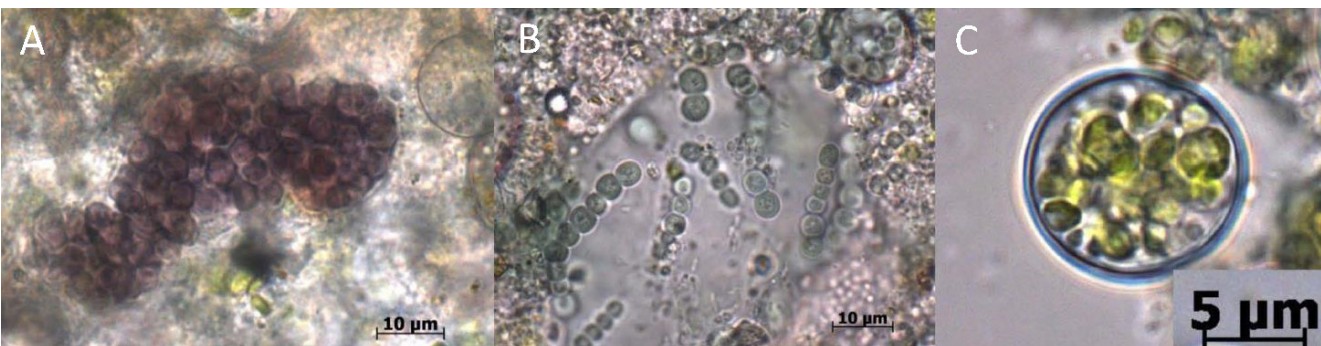

**Figure 6.** (**A**,**B**) *Nostoc* cf. *edaphicum*. (**C**) Sporocysts of *Myrmecia israelensis*. Scale bar: 10 μm in (**A**,**B**), and 5 μm in (**C**).

### 3.1.3. Sector T3

In sector T3, Sala del Aguila, the biofilms developed significantly. Phototrophic colonization adopts different morphologies in the distribution on the mineral substratum, from circular colonies to a homogeneous biofilm covering the rock surface.

The study of the cultured sample in the laboratory from Sala del Aguila biofilms showed cells of *Synechocystis* sp., *Diadesmis contenta*, *Phormidium* sp., *Chroococcidiopsis* sp., *Pseudococcomyxa simplex* and *M. israelensis*, as well as the bryophyte *E. verticillatum*.

In the samples from the circular colonies, a whitish mass formed by organisms that retained color, as well as many empty sheaths and dead organisms were revealed under the light microscope. This may be due to the alternation of dry and wet seasons.

We did not study fungi, because given the amount of bacteria and cyanobacteria in the biofilms, which usually produce bioactive (antifungal) substances, and in light of previous studies in other caves, it does not seem that fungi can attain great relevance in these phototrophic communities, as reported for Nerja Cave [15].

In the gallery accessing to Santuario de Noctiluca, the biofilms were also abundant. Under the light microscope, *M. israelensis* and *Phormidium* sp. were evidenced. The scanning electron microscope study showed a dense web of filaments of both cyanobacteria and associated filamentous bacteria.

The phototrophic organisms found in Tesoro Cave have been reported in many other European caves [14,37,41–49].

### 3.2. Preliminary Cleaning Testing

After reviewing the different cleaning procedures reported in the literature [50–53], we selected three methods: (i) mechanical cleaning with liquid nitrogen, (ii) cleaning with sodium hypochlorite, and (iii) cleaning with hydrogen peroxide. They were tested in biofilms from Sala del Aguila.

The cleaning with liquid nitrogen on biofilms was performed with a brush. Liquid nitrogen provided a cleaning technique that combined mechanical removal with freezing. This protocol, in the case of a biofilm composed mainly of the bryophyte *E. verticillatum*, or cyanobacteria and green algae, was much less effective than when using chemicals, because it was not able to completely remove the green biofilms.

The cleaning with sodium hypochlorite on the biofilm of cyanobacteria and algae was effective, and was maintained for a long time (Figure 7). This treatment has been applied in many caves all over the world for removing lampenflora [52]. Two months later, the rock presented a whiter hue than at the end of the cleaning, indicating that the chemical had a residual effect.

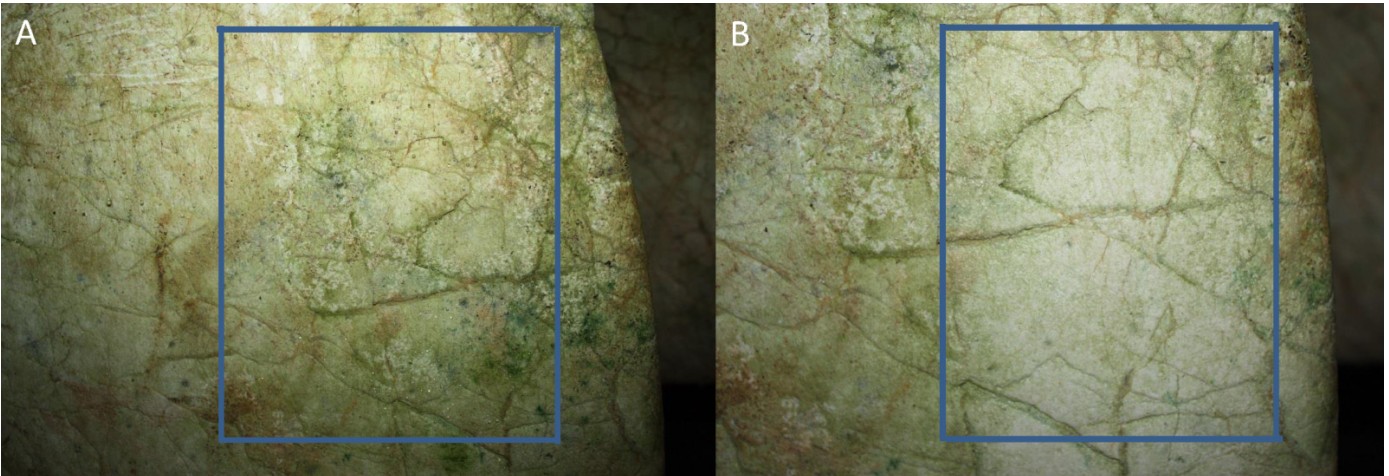

**Figure 7.** Cleaning of phototrophic biofilms with sodium hypochlorite. (**A**) Before treatment. (**B**) After treatment.

Cleaning with hydrogen peroxide on *E. verticillatum* showed that the hydrogen peroxide applied directly with a brush was effective. In the case of abundant colonization of bryophytes, it is preferable to use the mechanical removal of the organisms, that can be done easily, and to only apply hydrogen peroxide to the residual protonema and rhizoids that remain on the rock. Likewise, in the areas on which bryophytes are growing, lighting should be removed (Figure 8).

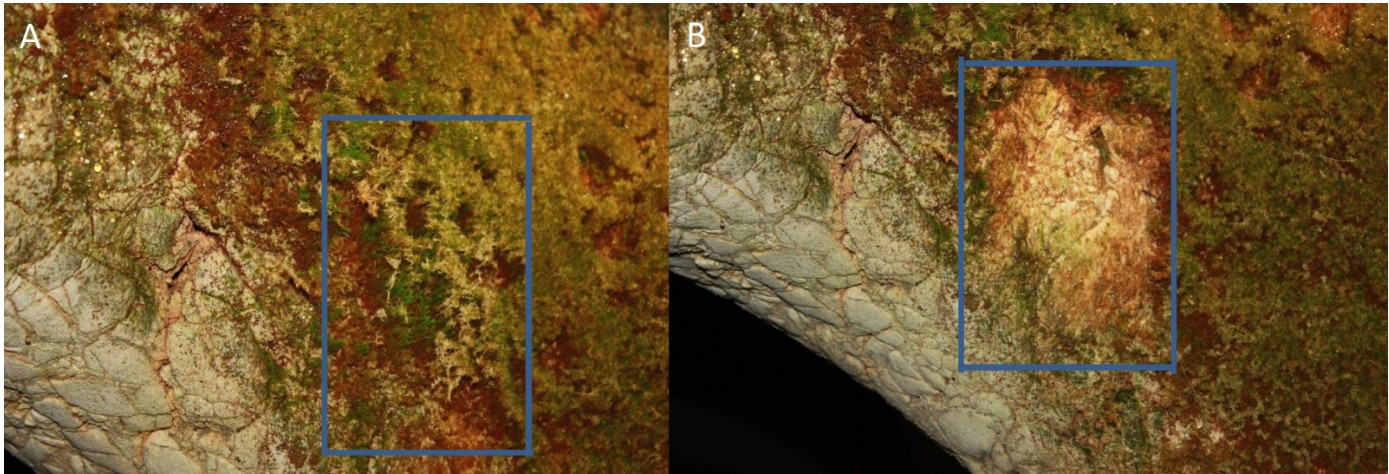

**Figure 8.** Cleaning with hydrogen peroxide on *Eucladium verticillatum.* (**A**) Before cleaning. (**B**) After cleaning.

The cleaning with hydrogen peroxide on biofilms of cyanobacteria, algae and bryophytes was very effective as it eliminated the phototrophic community without leaving residues (Figure 9). This was considered an environmentally acceptable procedure for removing cave biofilms [51]. Cleaning efficiency can be maintained after treatment if the lighting is removed after cleaning. In fact, under these environmental conditions, the cleaned areas should not develop new biofilms and will guarantee the persistence of the cleaning effect.

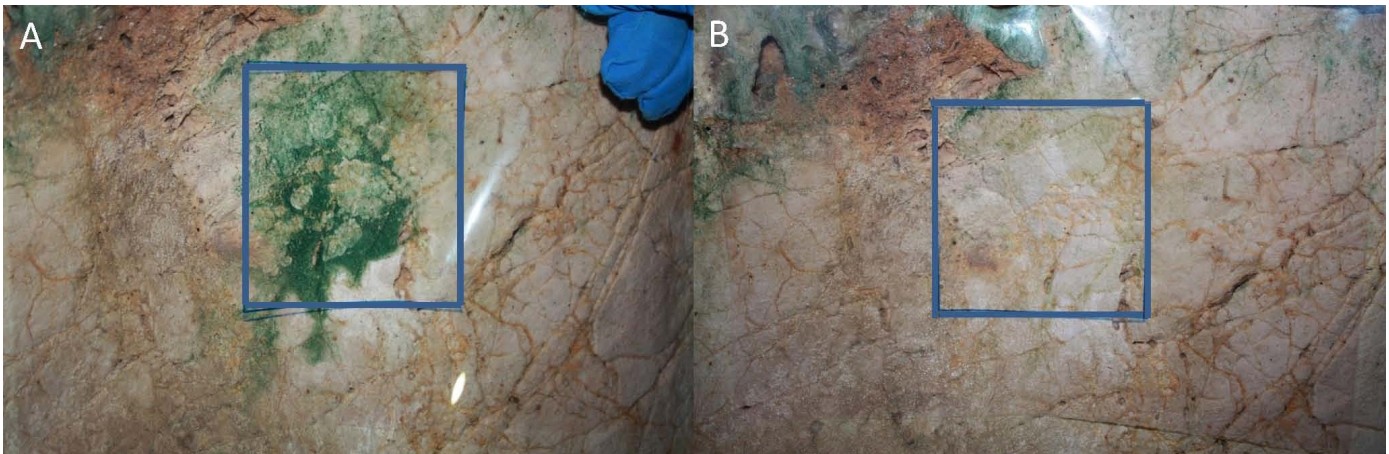

**Figure 9.** Cleaning of biofilms with hydrogen peroxide. (**A**) Before cleaning. (**B**) After cleaning.

From the preliminary tests carried out in the cave, it can be inferred that the most suitable methods for the cleaning and removal of the extensive green biofilms coating the cave surface were the use of hydrogen peroxide and sodium hypochlorite. The first should be used preferentially for its innocuousness. The second can leave residue, and odor can remain for a short time. Because of its complicated transport and execution, as well as possible danger to the applicator and for not having shown greater effectiveness than the other two treatments, the use of liquid nitrogen is not recommended.

### 3.3. Recommended Cleaning Protocols

The recommended procedure is first to carry out mechanical cleaning in those cases where the biofilm is thick, such as, for example, in cases of bryophytes and dense algae biofilms. Subsequently, hydrogen peroxide must be used to remove the adhered residues, and not be mechanically removed. This allows the use of less of the chemical and optimization of the results.

Cases where the green color is not completely eliminated can be treated exceptionally well with sodium hypochlorite, first testing a small rock area to check its effectiveness. If the green color does not disappear with the hydrogen peroxide, or subsequently with sodium hypochlorite, it is likely because the cells with the chlorophyll pigment are embedded in a calcite matrix.

Most of the rock art paintings are in a gallery not accessible to visitors (Galeria de Breuil, Figure 1), and therefore without lighting. No green biofilms were observed in this part of the cave. However, Sala del Aguila has a marked problem of colonization by phototrophic microorganisms, associated with the presence of a lighting source, which is very close to some of the rock art painting. This converts the cave into no longer a natural monument, but a site of cultural interest, and therefore the precautions that the current legislation provides for this site were adopted. The occurrence of a lighting point relatively near a painted horse protome, a typology from the Upper Paleolithic, threatened the painted area (Figure 10). The occurrence of bryophyte protonema and/or phototrophic microorganisms represents a real danger for the conservation of the painting. Considering these conditions, mechanical cleaning was proposed for the bryophytes and active phototrophic colonization, which was supervised at all times by the archaeological direction of the cave throughout the duration of the intervention.

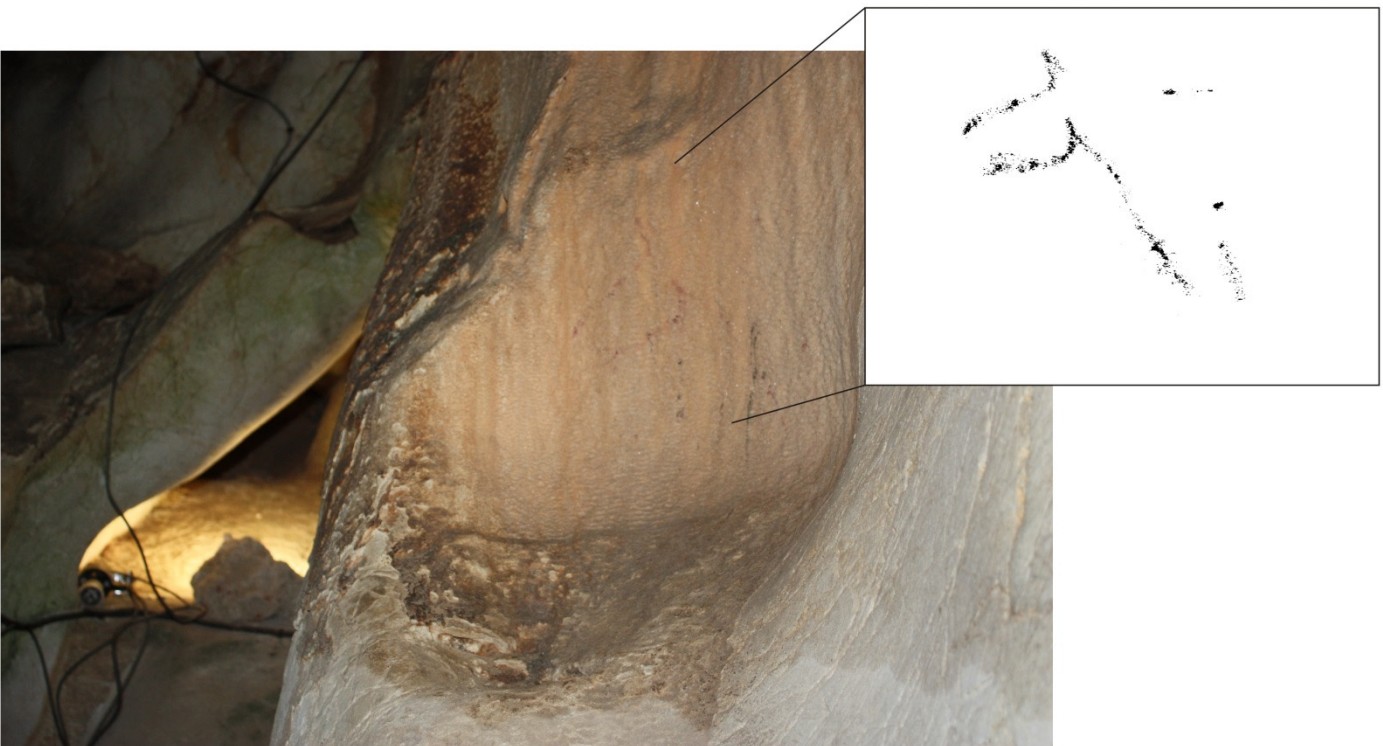

**Figure 10.** Lighting point in Sala del Aguila with phototrophic biofilms next to the Paleolithic painting. On the right, tracing of a Paleolithic horse protome using digital image analysis.

### 3.4. Effective Cleaning of Biofilms in Tesoro Cave by Agora S.L.

After the preliminary testing of the cleaning procedures, the treatments applied in cleaning the whole cave included mechanical removal and cleaning with hydrogen peroxide

and/or sodium hypochlorite. The distribution of cleaning treatments was as follows: physicochemical cleaning, 95% (hydrogen peroxide 94%, sodium hypochlorite 1%); only chemical cleaning, 5%: sodium hypochlorite. The products were applied according to the needs of the area to be cleaned. The cleaning lasted two months for three full-time restorers (Figure 11).

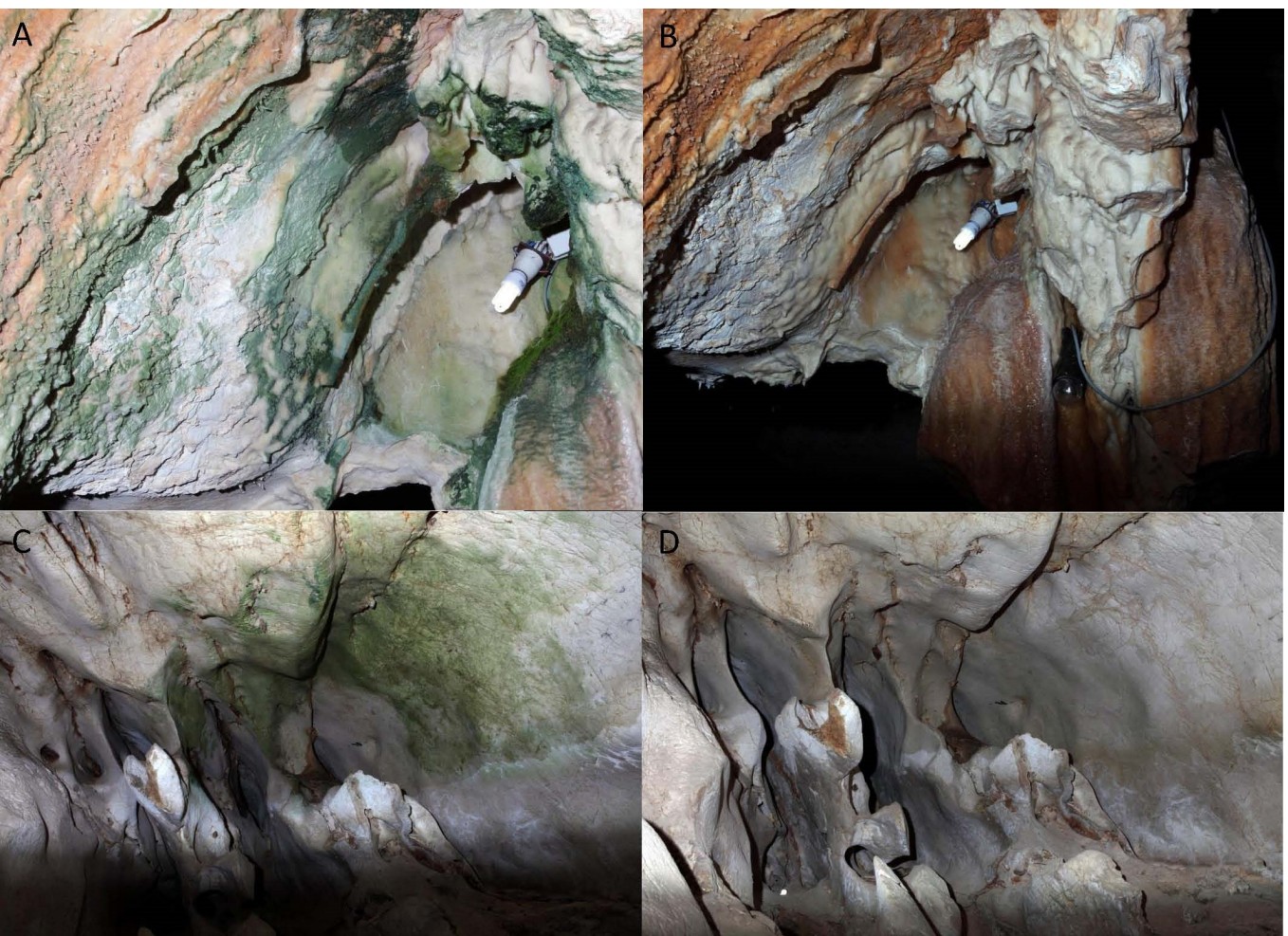

**Figure 11.** (**A**) Biofilms in Sala del Lago (T1) before cleaning. (**B**) After cleaning. (**C**) Biofilms in Sala del Aguila (T3) before cleaning. (**D**) After cleaning. Cleaning consisted of mechanical removal of biofilms and treatment of the residues with hydrogen peroxide.

## 4. Conclusions

The phototrophic biofilms covering the speleothems, walls and ground in Tesoro Cave, Rincon de la Victoria, Spain, were successfully cleaned using a protocol involving the mechanical removal of dense biomass, treatment with hydrogen peroxide and, when needed, it was followed by sodium hypochlorite. To guarantee long-term efficiency, the old lighting system should be replaced with a new design with monochrome or LED lamps to be located in the driest area of the cave, or alternatively, implementing visits with electric lanterns, which is possible due to the easy walkable cave trail.

**Author Contributions:** Conceptualization, C.S.-J.; investigation, V.J., M.H.-M., M.A.R.-C. and C.S.-J.; cleaning and restoration, F.R., C.A. and J.A.; writing—original draft preparation, C.S.-J.; writing—review and editing, C.S.-J. All authors have read and agreed to the published version of the manuscript.

**Funding:** The research and cave cleaning were funded by Rincon de la Victoria City Hall.

**Institutional Review Board Statement:** Not applicable.

**Informed Consent Statement:** Not applicable.

**Data Availability Statement:** Data supporting reported results on taxonomical and morphological identification of organisms are available from the authors.

**Acknowledgments:** The authors wish to acknowledge the professional support of the CSIC Interdisciplinary Thematic Platform Open Heritage: Research and Society (PTI-PAIS) as well as the facilities provided by Rincon de la Victoria City Hall for this study. The authors thank Jesus Muñoz Fuente, Real Jardin Botanico, CSIC, Madrid, for the identification of *Eucladium verticillatum*.

**Conflicts of Interest:** The authors declare no conflict of interest. The funders had no role in the design of the study; in the collection, analyses, or interpretation of data; in the writing of the manuscript, or in the decision to publish the results.

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
