# Peer review of "Cleaning of Phototrophic Biofilms in a Show Cave: The Case of Tesoro Cave, Spain"

_applsci, doi:10.3390/app12157357_

Round 1

Reviewer 1 Report

The article is very interesting and include very impressive photos.
I suggest the authors to discuss why certain bacterial and algae populations tend to develop biofilms and take over ecological niches in light exposure?

There is no data on the monitoring of the cleaning process by cultures; it only mentioned in the discussion that the rocks looked clean after two months. In order to prevent recurrence of the problem in the future, did the authors take cultures from the cleaned areas to test if something grew on the selective plates after the cleaning was completed?

The cleaning methods should be written in details in the Methods section (not only in the results section) and should include the concentration of the hydrogen peroxide and the sodium hypochlorite that were used for the cleaning. In addition, it should be noted how long the cleaning process lasted.  Since sodium hypochlorite can produce hydrochloric acid that can damaged the rocks. This possibility should be mention in the discussion.

Page 7 line 250 - the authors stated that they did not study fungi since it is not relevant to the phototrophic communities. Please discuss if fungi are relevant to form biofilm in the unilluminated area of the caves. Did fungi cause any damage to the rocks in these areas and what is known from the scientific literature?

Finally, please indicate if the chemical treatment had any effect on the rock art painting?

Author Response

REVIEWER 1

The article is very interesting and include very impressive photos.

I suggest the authors to discuss why certain bacterial and algae populations tend to develop biofilms and take over ecological niches in light exposure?

This was explained in lines  42-73 (always referring to the original manuscript).

There is no data on the monitoring of the cleaning process by cultures; it only mentioned in the discussion that the rocks looked clean after two months. In order to prevent recurrence of the problem in the future, did the authors take cultures from the cleaned areas to test if something grew on the selective plates after the cleaning was completed?

No samples from the cleaned area were taken, as the treatment with hydrogen peroxide and in minor percentage with sodium hypochlorite was expected to kill all microorganisms.

The cleaning methods should be written in details in the Methods section (not only in the results section) and should include the concentration of the hydrogen peroxide and the sodium hypochlorite that were used for the cleaning. In addition, it should be noted how long the cleaning process lasted.  Since sodium hypochlorite can produce hydrochloric acid that can damaged the rocks. This possibility should be mention in the discussion.

The cleaning of Tesoro Cave took two months for three fulltime restorers. The data were already included in the manuscript, Lines 330-337.

Page 7 line 250 - the authors stated that they did not study fungi since it is not relevant to the phototrophic communities. Please discuss if fungi are relevant to form biofilm in the unilluminated area of the caves. Did fungi cause any damage to the rocks in these areas and what is known from the scientific literature?

In lines 230-233 we wrote: “We have not studied fungi, because given the amount of bacteria and cyanobacteria in the biofilms, which usually produce bioactive (antifungal) substances, and in light of previous studies in other caves, it does not seem that fungi can attain great relevance in these phototrophic communities, as reported for Nerja Cave [15].”

In fact, in both Nerja and Tesoro caves, separated 50 km each other and suffering the same types of extensive biofilms, fungi were very scarce (see ref. 15 and list of eukaryotes). There, fungi represented 5% of the sequences in biofilms.

Finally, please indicate if the chemical treatment had any effect on the rock art painting?

Of course, no chemical treatment was applied to the rock art painting as the unique painting in the show area was not colonized by phototrophic microorganisms. Many other paintings were located in galleries not visited and without lighting. In addition, this treatment would be prohibited by the cultural authorities. However, the biofilms induced by a light and separated a few meters from the painting (see Figure 10, left side) were removed and the area cleaned.

Reviewer 2 Report

The article Cleaning of phototrophic biofilms in a show cave. The case of Tesoro Cave, Spain, about the problems generated in caves from artificial lighting for tourism and the methods to clean these surfaces with two different chemical methods is important and should be accepted. Nevertheless, there are important issues to attend:

There are repeated keywords from the title. The keywords are new opportunities for the document to be found by searching these words. If the words are already in the title, the authors are waisting opportunities of being found (and cited!) through different keywords. I suggest changing the repeated terms and writing other important words. 

L86: you might want to say RH, for relative humidity.

L88: please state first which are the normal values and who considers those values normal and why.

L89: What is an "anomalous concentration of CO2", why and according to whom under which reasons?

Please comment on how has this problem been addressed in the aforementioned caves (Lascaux, Altamira, etc.).

Materials and Methods

This part absolutely needs to be improved. Eppendorf tubes come in a variety of presentations and volumes. Were these sterile? How were they sterilized? 

L112: the authors isolated colonies from enrichment cultures. How were these cultures enriched? which culture media? the culture media was bought or prepared in situ? where did the nutrients come from? were these grown in flasks, tubes, or Petri dishes? for how long? how were these cultures grown? In 24 h light? which irradiance? temperature? 

L115: how were these samples prepared for SEM? which fixative was used? how were they dehydrated? for how long? were they dried using critical CO2 point or chemically? were they covered with gold? 

L120: Described elsewhere is not enough. Which primers did you use exactly? from where and why did you use those? Where did you buy them? Where did you buy all your chemicals and enzymes to work? 

L122: what do you mean by "uncultured clones"? If those are not cultured, how do you know they are clones?

L123: multiplied? The phylogenetic tree...

This section must be finished with all the details. The authors should never forget that the results should be repeatable so they have to provide ALL the information for other scientists to repeat the experiments.

Results

L133: high relative humidity gives no information. How high? compared to what?

L141: this bacterium (I think it is a bacterium??) has been also observed in other caves, but where are these caves located? Give at least some information about it.

Figure 2. Add the magnification used.

L166: Try not to start a sentence with an abbreviation.

Fig 3. Mark the strains used in this study in red or a different color.

L183: attached to

Fig 6. Magnification.

L240: most of this section belongs to material and methods.

Figs 7 and 8: mark with a square the cleaned area (like in Fig 9).

Fig 10: The tracing using digital image analysis is not clear. Tracing of what??

L316: most of this section belongs to Material and Methods

L343: After describing the cleaning with H2O2 and NaClO, which are chemicals, the authors add a "chemical cleaning", which is confusing.

Fig 11: add information about the cleaning method used on each surface.

Author contributions: delete the XX.

Author Response

REVIEWER 2

The article Cleaning of phototrophic biofilms in a show cave. The case of Tesoro Cave, Spain, about the problems generated in caves from artificial lighting for tourism and the methods to clean these surfaces with two different chemical methods is important and should be accepted. Nevertheless, there are important issues to attend:

There are repeated keywords from the title. The keywords are new opportunities for the document to be found by searching these words. If the words are already in the title, the authors are waisting opportunities of being found (and cited!) through different keywords. I suggest changing the repeated terms and writing other important words. 

Removed repeated keywords

L86: you might want to say RH, for relative humidity.

Corrected

L88: please state first which are the normal values and who considers those values normal and why.

Ranges reported in references 5-8 and 13-15. The ranges are normal for Spanish caves.

L89: What is an "anomalous concentration of CO2", why and according to whom under which reasons?

This is explained in the cited reference 16.

Please comment on how has this problem been addressed in the aforementioned caves (Lascaux, Altamira, etc.).

Commented

Materials and Methods

This part absolutely needs to be improved. Eppendorf tubes come in a variety of presentations and volumes. Were these sterile? How were they sterilized? 

Improved

L112: the authors isolated colonies from enrichment cultures. How were these cultures enriched? which culture media? the culture media was bought or prepared in situ? where did the nutrients come from? were these grown in flasks, tubes, or Petri dishes? for how long? how were these cultures grown? In 24 h light? which irradiance? temperature? 

Data included in the manuscript: Taxonomic identifications were based on morphological characters of the whole biofilms and on colonies grown on agar plates with media BG11 or BBM, incubated at room temperature and low light intensity [17,18].

L115: how were these samples prepared for SEM? which fixative was used? how were they dehydrated? for how long? were they dried using critical CO2 point or chemically? were they covered with gold? 

Reported in references added in the manuscript.

L120: Described elsewhere is not enough. Which primers did you use exactly? from where and why did you use those? Where did you buy them? Where did you buy all your chemicals and enzymes to work? 

All data reported in reference 14.

L122: what do you mean by "uncultured clones"? If those are not cultured, how do you know they are clones?

Corrected

L123: multiplied? The phylogenetic tree...

Corrected

This section must be finished with all the details. The authors should never forget that the results should be repeatable so they have to provide ALL the information for other scientists to repeat the experiments.

Results

L133: high relative humidity gives no information. How high? compared to what?

Near saturation, as the hall is occupied by a lake, as explained in the manuscript. Corrected.

L141: this bacterium (I think it is a bacterium??) has been also observed in other caves, but where are these caves located? Give at least some information about it.

Cyanidium is an alga (Rhodophyta). All requested data were reported in the original manuscript, lines 147-153 and references.

Figure 2. Add the magnification used.

Included in the photos.

L166: Try not to start a sentence with an abbreviation.

Corrected

Fig 3. Mark the strains used in this study in red or a different color.

Lines 126-127 reported the accession numbers of the sequences used, and these are included in Figure 3. In our opinion, this is clearly stated. No strains were used but clones to compare with strains previously isolated by other authors.

L183: attached to

Corrected

Fig 6. Magnification.

Included in the photos.

L240: most of this section belongs to material and methods.

We consider this section is Results for clarity purposes

Figs 7 and 8: mark with a square the cleaned area (like in Fig 9).

Marked

Fig 10: The tracing using digital image analysis is not clear. Tracing of what??

Added explanation

L316: most of this section belongs to Material and Methods

We consider this section is better in Results

L343: After describing the cleaning with H2O2 and NaClO, which are chemicals, the authors add a "chemical cleaning", which is confusing.

Corrected

Fig 11: add information about the cleaning method used on each surface.

Added

Author contributions: delete the XX.

Deleted

Reviewer 3 Report

This paper deals with the characterization of the phototropic community on the Tesoro Cave in Spain and the protocol used for its removal and both mechanical and chemical methods were used for this purpose.

I believe that the protocol used by the authors is not acceptable from an environmental point of view. In fact, as reported by Pérez [our Ref. 51], “Mechanical methods were also ruled out, given the sensitivity of some of the existing speleothems [mainly helictites] to scratching or roughing when brushes were used to eliminate algae that had calcified in many speleothems”.

As for chemical treatments, H2O2 can attack calcite of limestones and speleothems during spraying of lampenflora in cave environment [Faimon et al., 2003 – our Ref. 50]. To avoid the dissolution of calcite by the H2O2, presaturated with CaCO3 solutions need to be used. The authors used H2O2 dissolved in pure water and, in the more humid areas, it was applied even at 50%.

The use of hypochlorite solutions to clean show caves is not recommended for several reasons: (1) chlorine gas is released in the atmosphere and polluted the cave environment. (2) Chlorine decreases the pH and thus causes the dissolution and erosion of carbonate formations such as speleothems; (3) this biocide, applied as an aqueous solution, released some by-products on the surfaces leading to a degradative effect such as salt efflorescence or corrosion of the artefact surface (Warscheid and Braams, Int. Biodeter. Biodegr. 46, 343-368, 2000).

 The negative effects of the proposed treatments are shown in Figure 11 B, where the “Sala del Lago” site was reported after treatment and the formation of a yellow-brown patina is evident. This is a typical post-treatment degradation attributable to the application of high concentrations of oxidizing solutions; for instance, among oxidants, hypochlorite is known for its bleaching and yellowing properties.

Nowadays, several methods have been developed to reduce the drawbacks deriving from the use of oxidizing agents, encapsulating them in gel matrices (Gabriele et al., J. Cul. Herit. 49, 106–114, 2021; Gabriele et al., Int. Biodeter. Biodegr. 163, 105281, 2021) or replacing conventional biocides with essential oils (EOs) for the restoration of cultural heritage, as reported in these two reviews (Fidanza and Caneva, J. Cul. Herit. 38, 271–286, 2019; Cappitelli et al., Microorganisms, 8, 1542, 2020). EOs were successfully used to prevent biodeterioration of the Catacombs of SS. Marcellino and Pietro, in Rome (Italy) (Bruno et al., Ann. Microbiol. 69, 1023–1032, 2019). On the other hand, more recently, Argyri and coworkers (Microorganisms, 9, 1836, 2021) tested a series of 18 essential oils (EOs) as antimicrobial agents towards several microorganisms (bacteria and fungi) isolated from Petralona Cave in Greece. They demonstrated that EOs have the potential to be applied as biocides to address the serious issue of interior alteration observed in show caves.

For all these reasons, I don’t think this paper is suitable for publication in Applied Sciences.

Author Response

REVIEWER 3

This paper deals with the characterization of the phototropic community on the Tesoro Cave in Spain and the protocol used for its removal and both mechanical and chemical methods were used for this purpose.

I believe that the protocol used by the authors is not acceptable from an environmental point of view. In fact, as reported by Pérez [our Ref. 51], “Mechanical methods were also ruled out, given the sensitivity of some of the existing speleothems [mainly helictites] to scratching or roughing when brushes were used to eliminate algae that had calcified in many speleothems”.

As for chemical treatments, H2O2 can attack calcite of limestones and speleothems during spraying of lampenflora in cave environment [Faimon et al., 2003 – our Ref. 50]. To avoid the dissolution of calcite by the H2O2, presaturated with CaCO3 solutions need to be used. The authors used H2O2 dissolved in pure water and, in the more humid areas, it was applied even at 50%.

The use of hypochlorite solutions to clean show caves is not recommended for several reasons: (1) chlorine gas is released in the atmosphere and polluted the cave environment. (2) Chlorine decreases the pH and thus causes the dissolution and erosion of carbonate formations such as speleothems; (3) this biocide, applied as an aqueous solution, released some by-products on the surfaces leading to a degradative effect such as salt efflorescence or corrosion of the artefact surface (Warscheid and Braams, Int. Biodeter. Biodegr. 46, 343-368, 2000).

 The negative effects of the proposed treatments are shown in Figure 11 B, where the “Sala del Lago” site was reported after treatment and the formation of a yellow-brown patina is evident. This is a typical post-treatment degradation attributable to the application of high concentrations of oxidizing solutions; for instance, among oxidants, hypochlorite is known for its bleaching and yellowing properties.

Nowadays, several methods have been developed to reduce the drawbacks deriving from the use of oxidizing agents, encapsulating them in gel matrices (Gabriele et al., J. Cul. Herit. 49, 106–114, 2021; Gabriele et al., Int. Biodeter. Biodegr. 163, 105281, 2021) or replacing conventional biocides with essential oils (EOs) for the restoration of cultural heritage, as reported in these two reviews (Fidanza and Caneva, J. Cul. Herit. 38, 271–286, 2019; Cappitelli et al., Microorganisms, 8, 1542, 2020). EOs were successfully used to prevent biodeterioration of the Catacombs of SS. Marcellino and Pietro, in Rome (Italy) (Bruno et al., Ann. Microbiol. 69, 1023–1032, 2019). On the other hand, more recently, Argyri and coworkers (Microorganisms, 9, 1836, 2021) tested a series of 18 essential oils (EOs) as antimicrobial agents towards several microorganisms (bacteria and fungi) isolated from Petralona Cave in Greece. They demonstrated that EOs have the potential to be applied as biocides to address the serious issue of interior alteration observed in show caves.

For all these reasons, I don’t think this paper is suitable for publication in Applied Sciences.

Responses:

As scientists deeply involved in the conservation of cultural heritage we are aware of all methods (friendly and non-friendly) to be used. We also tested the efficiency of natural biocides on stone in a previous paper: Sasso et al. (2016) Potential of natural biocides for biocontrolling phototrophic colonization on limestone. Int. Biodeter. Biodegr. 107, 102-110.

Biofilms on cultural heritage stone are formed by complex microbial communities where bacteria, cyanobacteria, algae, fungi, lichens, bryophytes, protozoa, rotifers, ciliates, nematodes, etc., are involved and interact (e.g. Grbic et al. 2010, Arch. Biol. Sci., Belgrade, 62, 625; Polo et al. 2012, Biofouling 28, 1093; Urzi et al. 2016; Sci. Total Environ. 572, 252; Gallego-Cartagena et al. 2020, Int. Biodeter. Biodegr. 147, 104874; Ding et al., 2021, Corros. Mater. Degrad. 2, 31; Jurado et al. 2020, Appl. Sci. 10, 3448), therefore EOs or natural biocides targeting a specific group of microorganisms is not useful and its application is unrealistic.

In addition, there is a high number of papers reporting the inhibition in the laboratory of selected compounds on a single bacterium, fungus, or alga, however the application of such compounds in the field (monuments, subterranean environments, etc.) is rarely reported because several bias, including the great extension of rock or stone to be treated, the selective action on a determined group of microorganisms, but not on all those forming the complex biofilms, and third, that restoration companies do not include EOs or natural biocides in their portfolio, due to the difficult or impossible application in the field (large surface extensions, time-consuming protocols, etc.) of previously tested laboratory assays in small stone pieces.

As a matter of fact and example, a few fungal metabolites have been considered as potential antifouling agents for stone cleaning (Petraretti et al. 2022, Toxins 14, 407). These compounds: cavoxin, epi-epoformin, seiridin and sphaeropsidone were studied 40 years ago (Evidente et al. 1985, J. Nat. Comp. 48, 916; Evidente and Randazzo, 1986, J. Nat. Comp. 49, 593), and their properties were tested.

As far as known, and as reported in the literature, most of these metabolites have not been tested as antifouling (fouling is the accumulation of biofilms of microorganisms on surfaces), or on natural biofilms, but only on single fungal species in the laboratory or with stones inoculated with fungal species (e.g. Evidente et al. 1998, Phytochemistry 48, 1139; Schrader et al. 2010, Chem. Biodiver. 7, 2261; Evidente et al., 2011, J. Nat. Comp. 74, 757; Masi et al., 2021, Biomolecules 11, 295, etc.). This is an example of the bias of the methodology and how the authors try to extrapolate partial results obtained in the laboratory to the field and to nature, which is not acceptable. An antifouling compound should be active against all members of the complex biofilm, not only on one that might be the less important in the microbial consortium.

Questions such as: why these fungal metabolites are not widely used at present when they were known about 40 years ago? What validity has, therefore, all the studies carried out along the time? What problems: commercial, low or no activity against a wide spectrum of microorganisms (bacteria, cyanobacteria, algae, fungi, etc., which are components of most biofilms), etc. prevented their use in cultural heritage? Are the restoration companies prepared for the application of friendly or ecological cleaning treatments on extensive surfaces? Unfortunately, the response is NO.

Similar trends occur with EOs. Most of the paper found in the literature deal with the antimicrobial (bacteria and fungi) effect of EOs (Solorzano-Santos and Miranda-Novales, 2012, Curr. Op. Biotechnol. 23, 136-141; Aelenei et al. 2016, Medicines 3, 19; Chouhan et al. 2017, Medicines 4, 58; Kon et al., 2014, Exp. Rev. Anti-infect. Therapy 10, 775-790; Ribeiro et al., 2020, Antibiotics 9, 0717; etc.).  What about algae, lichens, mosses, amebae, and other microscopic eukaryotes, etc.? These organisms were rarely or never considered in the published studies.

The reviewer should accept that although there are several reports on the efficacy of EOs in cultural heritage, always tested in the laboratory, and on specific microorganisms, they do not cover the wide spectrum of micro- and macro-organisms composing the cave biofilms.

Bruno et al., 2019, Ann. Microbiol. 69, 1023–1032, testing the application of EOs on agar-cultured phototrophic biofilms, composed of species of cyanobacteria isolated previously from different Roman catacombs, wrote: “Along with phototrophs, heterotrophic microorganisms are major players in the formation of these communities; thus, future consideration of community successions and interactions will need to be addressed to fully understand the overall biofilm dynamics, as well as the need to characterise the onset and progression of biofilm formation; follow-up studies will be needed to assess the efficacy of the treatment used and to plan more effective conservative management strategies”.

Ranaldi et al., 2022, Int. Biodeter. Biodegr. 172, 105436, tested EOs suspended into an alginate hydrogel.  Five cyanobacteria were tested in the laboratory and showed membrane disruption, but not data on the effect on other biofilms microorganisms was reported.

Other papers mentioned by the reviewer: (Gabriele et al., J. Cul. Herit. 49, 106–114, 2021; Gabriele et al., Int. Biodeter. Biodegr. 163, 105281, 2021) tested an alginate hydrogel-chlorine in the laboratory and in a small area of a church. While the application of such hydrogels is feasible in small areas, it hardly can be used in a cave with hundreds to thousands m2 of rocks coated by biofilms.

Application of alginate to cultural heritage poses a problem that was not discussed by the authors suggesting the use of such hydrogels. In fact, alginate is a linear polysaccharide that can be degraded by microorganisms (Carwell, 1988, PhD thesis, Univ of Wales; Min et al. 1977, Biochem J. 81, 555-562; Schaumann and Weide, 1990, Hydrobiologia 204/205, 589-596; Cruz et al., 2013, Appl. Microbiol. Biotechnol. 97, 9847-9858; Bai et al, 2017, Plos One 12, e0171576), and particularly soil microorganisms such as Flavobacterium, Alkaligenes, Bacillus, Sphingomonas, Pseudomonas, etc. widely retrieved from caves (Bashan, 1986, Appl. Environ. Microbiol. 51, 1089-1098; Hashimoto et al., 2000, J. Bacteriol 182, 4572-4577; Kaneko et al. 2000, J. Ferm. Bioeng. 69, 192-194; Xiao et al., 2006, World J. Microbiol. Biotechnol. 22, 81-88; Phang et al., 2011, Polymer  Degrad. Stabil. 96, 1653-1661; Subaryono et al., 2013, Squalen Bull. Mar. Fish. Postharv. Biotechnol. 8, 105-116; Tavafi et al., 2017, Iran Biomed. J. 21, 48-56), therefore, the residues left by their application could be dangerous for the cultural properties as they can be used as carbon source for secondary invasions of microorganisms colonizing the cleaned surface.

Regarding the mention to Argyri et al. 2021, Microorganisms, 9, 1836, they studied 18 essential oils (EO) of plant origin extracted from various Greek plants and evaluated their antimicrobial activity against 35 bacterial and 31 fungi isolates (isolated from a Greek cave). No assays were carried out on cyanobacteria,algae, rhodophyta, etc. that are the main components of green biofilms in cave and in Tesoro Cave. Therefore this paper cannot be claimed in support of an application to the cave lampenflora.  In addition, the authors cleary stated that “It has to be noted that the effectiveness of the selected EOs studied in this work has to be evaluated in situ, by applying the final product in real cave ecosystems.” In general, most authors, as Argyri et al., only tested EOs on bacteria and fungi (see Argyri discussion), but not on phototrophic microorganisms and eukaryotes, some of the main components of cave biofilms.

In addition, we doubt that EOs will kill all biofilm microorganisms, and some microorganisms will used EOs as carbon source. The literature is full of reports on the biodegradation of terpenoids and related compounds (EOs) by bacteria in soils (Hunt et al. 1980, Nature 288, 577-578; Mikami, 1988, Biotecnol. Gen. Eng.Rev.6, 271-320; Adams et al. 2013, Appl. Environ. Microbiol. 79, 3468-3475; Marmulla and Hardr, 2014, Front. Microbiol. 5, 346; Vilanova et al. 2014, Plos ONE 9, e100740; Drummond 2022, Chem. Biodiver. 19, e202100734, just for citing a few of them).

Regarding the reviewer quotation: “…. this biocide (chlorine), applied as an aqueous solution, released some by-products on the surfaces leading to a degradative effect such as salt efflorescence or corrosion of the artefact surface (Warscheid and Braams, Int. Biodeter. Biodegr. 46, 343-368, 2000).” In the Warscheid paper, the unique reference to chlorine is: “Other bleaching agents, like hydrogen peroxide, chlorine and chloramine, are also unsuitable in conservation practice because they may oxidize iron inclusions in the mineral material (Kumar and Kumar, 1999).” However, the sentence wrote by the reviewer can be found in the paper of Gabriele et al., 2021, Int. Biodeter. Biodegr. 163, 105281: “In particular, when these biocides are applied as an aqueous solution, some by-products remain on the surfaces leading to a degradative effect such as salt efflorescence or corrosion of the artefact surface (Pozo-Antonio et al., 2016).”

To conclude, in the paper was clearly stated that 94% of the surface was cleaned with hydrogen peroxide, with a subsequent treatment with sodium hypochlorite only in 1% of the cases. The remaining 5% was cleaned with sodium hypochlorite in areas where the biofilms were entrapped into a calcite layer and in sandy surfaces with little physical compaction. This represented 10 m2 of the entire treated surface.

The reviewer claims that the protocol used by the authors is not acceptable from an environmental point of view. However, the protocol is operative from a practical point of view because no environmental protocol (EOs), as suggested, can be applied to the cave. While the application of such hydrogels and EOs is feasible in small areas, mainly in laboratory essays, it cannot be applied in a cave with more than two hundred m2 of rocks coated by biofilms to be removed and cleaned. The budget of the cave owner cannot cover the restorators manpower time needed to clean the cave with this methodology.

The reviewer stated: The negative effects of the proposed treatments are shown in Figure 11 B, where the “Sala del Lago” site was reported after treatment and the formation of a yellow-brown patina is evident. This is a typical post-treatment degradation attributable to the application of high concentrations of oxidizing solutions; for instance, among oxidants, hypochlorite is known for its bleaching and yellowing properties.

We are sorry to disagree with this statement. Dark to brown speleothems are quite common in caves and mines due to iron, manganese and other mineral deposits. In fact, calcite speleothems can be white, light brown, brown or dark in color (see: Sawlowicz et al., 2014, Geol. Quarter. 58, 449-458; Gázquez et al. 2013, Geomorphology 198, 138-146; Pontes et al., 2020, Int. J. Speleol. 49, 119-136). The reviewer can clearly see in Figure 11 A that the yellow-brown color is natural to the rock, as denoted from the areas not covered by biofilms (see figure and arrow). Anyway, why this claimed “post-treatment degradation attributable to the application of high concentrations of oxidizing solutions” cannot be seen in Figure 11 D, subjected to the same treatment and where the rocks are white?

Maybe the reviewer is influenced by data published on EOs and other recent treatments mainly based on laboratory assays, but these treatments will take a long time to become a common practice in restoration companies, if finally adopted.

At present, the cleaning of extensive biofilms covering hundreds to thousands m2 in caves and stone surfaces should rely on methods affordable by the restoration companies and approved by the cultural authorities, as it was in this case.

The cleaning of Tesoro Cave took two months for three fulltime restorers. How long will take a cleaning with other less fast and time-consuming procedures, as EOs? It is affordable economically?

Finally, as example of the difficult implementation of EOs in larger surfaces is the work of Spada et al. 2021, Int. Biodeter. Biodegr. 163, 105280, which is in favour of our opinions. The authors tested 10 different EOs using three different carriers in the laboratory and selected the best EOs. When applied in situ, their conclusions were: “To obtain a complete cleaning of the statue we had to carry out four applications and this amplified the running time of the work (about two months for the entire sculpture)”, with five scientists.” How much time and people in the case of a cave?

We believe that our data, and many others, support the use of hydrogen peroxide for cleaning almost the entire cave (94%), and that the cleaning of residual and resistant biofilms support the benefit of using sodium chloride in a very small extent (10 m2), in spite that some calcite surfaces can be attacked. Please, take into account that calcite from speleothems is also continuously corroded by condensation water and CO2 from visitors. This is a matter widely discussed in the literature (Sarbu and Lascu, 1997, J. Cave Karst Stud. 59, 99-102; Baker and  Genty, 1998, J. Environ. Manag. 55, 165-175; Sanchez-Moral et al. 1999, Sci. Total Environ. 243-244, 67-84; Faimon et al. 2006, Sci. Total Environ. 369, 231-245; Martin-Garcia et al. 2011, Carbon. Evapor. 26, 83-94). Therefore, the possible corrosion induced by the hypochlorite in a small cave extension is likely small compared with the continuous corrosion induced by condensation water and CO2 along the years by thousands of visitors.

Round 2

Reviewer 2 Report

After using my night time reviewing this paper again, I am wondering why the editorial is even asking for peer review if the authors will decide not to attend the recommendations.

The authors did not provide all the relevant information they were asked to provide. This paper cannot be published with all this missing information.

I suggest that the authors clarify the material and methods section with the relevant information that is missing, including that in the results section that does not belong there.

Comments:

L89: What is an "anomalous concentration of CO2", why and according to whom under which reasons?

This is explained in the cited reference 16.

The authors added references, but still not clear what is an anomalous concentration of CO2. Yes, the reference is there, but the authors are supposed to give all the important information to the reader, not only the reference for the reader to search for the information. Please, state how anomalous this gas was.

Materials and Methods

This part absolutely needs to be improved. Eppendorf tubes come in a variety of presentations and volumes. Were these sterile? How were they sterilized? 

Improved

The authors responded partially. I will repeat my question: how were these tubes sterilized? Eppendorf tubes come in various presentations and volumes, please specify the volume.

 L112: the authors isolated colonies from enrichment cultures. How were these cultures enriched? which culture media? the culture media was bought or prepared in situ? where did the nutrients come from? were these grown in flasks, tubes, or Petri dishes? for how long? how were these cultures grown? In 24 h light? which irradiance? temperature? 

Data included in the manuscript: Taxonomic identifications were based on morphological characters of the whole biofilms and on colonies grown on agar plates with media BG11 or BBM, incubated at room temperature and low light intensity [17,18].

Again, incomplete information. Agar plates with ?? % of agar. With BG11 or BBM medium bought already prepared? or prepared at the laboratory? Which nutrients (trademarks, purity) did you use? Please, don’t be scarce with the information. Remember that materials and methods are for the reader to be capable of reproducing all your experiments!

Please, read again my original questions and answer all of them.

Low light intensity: I know it is not easy to be measured, but now there are even smartphone apps that can do it.

L115: how were these samples prepared for SEM? which fixative was used? how were they dehydrated? for how long? were they dried using critical CO2 point or chemically? were they covered with gold? 

Reported in references added in the manuscript.

This information is still not there. Please, do not add only a reference; explain in a few lines how you prepared the samples. This is of great importance. It is not necessary that you repeat all the methods, but a summary.

Mention the clones’ names. This is vital to understanding the phylogenetic tree.

L120: Described elsewhere is not enough. Which primers did you use exactly? from where and why did you use those? Where did you buy them? Where did you buy all your chemicals and enzymes to work? 

All data reported in reference 14.

Still not enough to report only a reference. Authors MUST add at least some information on this too.

L122: what do you mean by "uncultured clones"? If those are not cultured, how do you know they are clones?

Corrected

This was not corrected, only erased “uncultured.” So maybe you worked only with cultured clones? If so, please specify which ones and how many.

L123: multiplied? The phylogenetic tree...

Corrected

This section must be finished with all the details. The authors should never forget that the results should be repeatable, so they must provide ALL the information for other scientists to repeat the experiments.

L141: this bacterium (I think it is a bacterium??) has been also observed in other caves, but where are these caves located? Give at least some information about it.

Cyanidium is an alga (Rhodophyta). All requested data were reported in the original manuscript, lines 147-153 and references.

The authors MUST provide the whole information and not only references. We are not asking for nonsense, only to mention where is the Nerja Cave located. Please, provide this information in the manuscript.

Figure 2. Add the magnification used.

Included in the photos.

No. Magnification is not included in the pictures. A scale bar is. But you must include the objective you were using (60x? 100x?) and multiply it by the ocular magnification (10 or 20), so that you must state “Magnification 1000x”.

Please add the magnification used. Write this information in the figure caption.

Fig 3. Mark the strains used in this study in red or a different color.

Lines 126-127 reported the accession numbers of the sequences used, and these are included in Figure 3. In our opinion, this is clearly stated. No strains were used but clones to compare with strains previously isolated by other authors.

Please make it easier to find the clones in the tree.

Fig 6. Magnification.

Included in the photos.

No. Magnification is not included in the pictures. A scale bar is. But you must include the objective you were using (60x? 100x?) and multiply it by the ocular magnification (10 or 20), so that you must state “Magnification 1000x”.

Please add the magnification used. Write this information in the figure caption.

L240: most of this section belongs to material and methods.

We consider this section is Results for clarity purposes

It is not clear if not in the material and methods section.

L316: most of this section belongs to Material and Methods

We consider this section is better in Results

These are NOT results. they are MIXED. This information does not belong here. Must be stated in M&M. And the results, in results section.

Author Response

Second revision Tesoro

Reviewer 2

After using my night time reviewing this paper again, I am wondering why the editorial is even asking for peer review if the authors will decide not to attend the recommendations.

The authors did not provide all the relevant information they were asked to provide. This paper cannot be published with all this missing information.

I suggest that the authors clarify the material and methods section with the relevant information that is missing, including that in the results section that does not belong there.

Data from the Results sections moved to Material and Methods, see lines 150-180.

Comments:

L89: What is an "anomalous concentration of CO2", why and according to whom under which reasons?

20.000 ppm. Data added in the second revision (lines 92-96) as well as a paragraph explaining the environmental conditions.

The authors added references, but still not clear what is an anomalous concentration of CO2. Yes, the reference is there, but the authors are supposed to give all the important information to the reader, not only the reference for the reader to search for the information. Please, state how anomalous this gas was.

Materials and Methods

This part absolutely needs to be improved. Eppendorf tubes come in a variety of presentations and volumes. Were these sterile? How were they sterilized? 

The authors responded partially. I will repeat my question: how were these tubes sterilized? Eppendorf tubes come in various presentations and volumes, please specify the volume.

Sterile Eppendorf tubes were supplied by a company. The tube volume has been included in the text (lines 117-118).

 L112: the authors isolated colonies from enrichment cultures. How were these cultures enriched? which culture media? the culture media was bought or prepared in situ? where did the nutrients come from? were these grown in flasks, tubes, or Petri dishes? for how long? how were these cultures grown? In 24 h light? which irradiance? temperature? 

Data included in the manuscript: Taxonomic identifications were based on morphological characters of the whole biofilms and on colonies grown on agar plates with media BG11 or BBM, incubated at room temperature and low light intensity [17,18].

Again, incomplete information. Agar plates with ?? % of agar. With BG11 or BBM medium bought already prepared? or prepared at the laboratory? Which nutrients (trademarks, purity) did you use? Please, don’t be scarce with the information. Remember that materials and methods are for the reader to be capable of reproducing all your experiments!

Both media, BG11 and BBM, were from Sigma-Aldrich. This information has been included in the text as well as the percentage of agar. Lines 120-123)

Sorry, the readers cannot be capable of reproducing all the experiments, which require access to Tesoro Cave, and are unable to sampling on biofilms, which were removed and cleaned.

Please, read again my original questions and answer all of them.

Low light intensity: I know it is not easy to be measured, but now there are even smartphone apps that can do it.

Culture conditions explained in the manuscript. No measurements at such stage. Lines 122-123.

L115: how were these samples prepared for SEM? which fixative was used? how were they dehydrated? for how long? were they dried using critical CO2 point or chemically? were they covered with gold? 

This information is still not there. Please, do not add only a reference; explain in a few lines how you prepared the samples. This is of great importance. It is not necessary that you repeat all the methods, but a summary.

Data added in the revised manuscript. Lines 147-149.

Mention the clones’ names. This is vital to understanding the phylogenetic tree.

L120: Described elsewhere is not enough. Which primers did you use exactly? from where and why did you use those? Where did you buy them? Where did you buy all your chemicals and enzymes to work? 

All data reported in reference 14.

Still not enough to report only a reference. Authors MUST add at least some information on this too.

The requested information was included in the lines 125-136. Figure legend modify for clarification.

L122: what do you mean by "uncultured clones"? If those are not cultured, how do you know they are clones?

Corrected

This was not corrected, only erased “uncultured.” So maybe you worked only with cultured clones? If so, please specify which ones and how many.

The protocol for cloning has been included. “DNA libraries of PCR amplified products were constructed using the TOPO-TA cloning kit (Invitrogen, Carlsbad, California, USA) according to the manufacture´s recommendations. Plasmids were extracted with the JetQuick Plasmid Miniprep Spin kit (Genomed, Löhne, Germany)…” Lines 131-136.

L123: multiplied? The phylogenetic tree...

Corrected

This section must be finished with all the details. The authors should never forget that the results should be repeatable, so they must provide ALL the information for other scientists to repeat the experiments.

Sorry, the readers cannot be capable of reproducing all the experiments, which require access to Tesoro Cave, and are unable to sampling on biofilms, which were removed and cleaned.

L141: this bacterium (I think it is a bacterium??) has been also observed in other caves, but where are these caves located? Give at least some information about it.

The authors MUST provide the whole information and not only references. We are not asking for nonsense, only to mention where is the Nerja Cave located. Please, provide this information in the manuscript.

Cyanidium is an alga (Rhodophyta). It was cleary stated in the abstract and though the manuscript.  The requested information on location of Nerja Cave in line 194.

Figure 2. Add the magnification used.

Included in the photos.

No. Magnification is not included in the pictures. A scale bar is. But you must include the objective you were using (60x? 100x?) and multiply it by the ocular magnification (10 or 20), so that you must state “Magnification 1000x”.

Sorry. No need to provide such information on magnification. This is not the usual way in the papers. All published papers only report scale bar, which is enough to known the alga sizes. A few examples were selected:

In Albertano et al. 2005. New strategies for the monitoring and control of cyanobacterial films on valuable lithic faces, Plant Biosystems 139, 311-322,

Figure 3. Two sampling sites at the Catacombs of Domitilla and St. Callistus in Rome (a, b), stereoview (c – f) and light-micrographs (g, h) of sampled biofilms. (a) Detail of one arcosolium inside the ‘‘Cubicolo degli Apostoli Piccoli’’ at the Catacomb of Domitilla. Blue–green

biofilms are visible on the bottom right. (b) Frescoes inside the ‘‘Cubicolo di Oceano’’ at the Catacomb of St. Callistus with spotted phototrophic growth. (c) Pale-green patina on a plaster slide formed by Chroococcalean cyanobacteria, actinobacteria and Scytonema julianum. (d) S. julianum, Fischerella sp. and Phormidium sp. biofilm. (e) Detail of Scytonema julianum calcified filaments on a stone surface and (f) inside a micro-cavity. (g) Disaggregated biofilm dominated by Phormidium sp. in which chroococcal cyanobacteria are also visible. (h) Calcified S. julianum filaments and Fischerella sp. Scale bar 10 mm.

In Bruno et al. 2019. Biodeterioration of Roman hypogea: the case study of the Catacombs of SS. Marcellino and Pietro (Rome, Italy). Annals of Microbiology 69, 1023–1032

Fig. 3 Microscopic observation of biofilm samples collected in situ: a and b micrographs of samples from site 1. Scale bar = 10 μm. c and d Micrographs of samples from site 2. Scale bar = 10 μm. e Sample from site 1 observed at the CLSM showing heterocytous cyanobacteria belonging to the genera Symphyonemopsis and Scytonema julianum. Scale bar = 20 μm. f Sample from site 2 observed at the CLSM showing filaments of Scytonema julianum with exopolysaccaridic envelopes (green) along with Symphyonemopsis sp. and the small filaments of Oculatella subterranean. Scale bar=20 μm.

In Lamprinou et al. 2011. Morphology and molecular evaluation of Iphinoe spelaeobios gen. nov., sp. nov. and Loriellopsis cavernicola gen. nov., sp. nov., two stigonematalean cyanobacteria from Greek and Spanish caves. International Journal of Systematic and Evolutionary Microbiology 61, 2907–2915.

Fig. 1. LM images of Iphinoe spelaeobios gen. nov., sp. nov. showing filaments and type of branching (a–d), hormocyst with terminal heterocyst  (e), filament with evidence of the presence of false branching (f) and filament with intercalary heterocyst (g). Bars, 10 mm.

In Lamprinou et al. 2011. Morphology and molecular evaluation of Iphinoe spelaeobios gen. nov., sp. nov. and Loriellopsis cavernicola gen. nov., sp. nov., two stigonematalean cyanobacteria from Greek and Spanish caves. International Journal of Systematic and Evolutionary Microbiology 61, 2907–2915.

Fig. 2. Micrographs of Iphinoe spelaeobios gen. nov., sp. nov. obtained by TEM (a–g) and SEM (h–k): (a) longitudinal section showing thylakoids occupying the whole cell area; (b) section of a filament showing trichome disintegration with the help of a necridic cell (NC), accompanied by mucilaginous biconcave lamellae formation (BL); (c, d) sections of filaments indicating generation of T type of branching; (e–g) septum without intercellular connections between vegetative cells; (h–k) calcified sheaths, cylindrical or torulose filaments and type of branching. Bars: (a–d) 2 mm; (e) 0.5 mm; (f, g) 0.2 mm; (h–k) 10 mm.

Please add the magnification used. Write this information in the figure caption.

Scale bar was added in all figure legends.

Fig 3. Mark the strains used in this study in red or a different color.

Please make it easier to find the clones in the tree.

Lines 126-127 reported the accession numbers of the sequences used, and these are included in Figure 3. In our opinion, this is clearly stated. No strains were used but clones to compare with strains previously isolated by other authors. There is no way to confuse clones with their accession numbers with strains, as the isolated strains are reported with full name and isolation location.

See the manuscript paragraphs where this is clearly stated:

The first morphospecies of the genus Cyanidium that did not inhabit acidic ecosystems or volcanic areas were described from samples from caves (Cyanidium chilense) and from fissures in coastal rocks, both in Chile [32]. Subsequently, by means of molecular analysis, three species of non-extremophilic aerophytic Cyanidium have been described, two of them Cyanidium sp. Monte Rotaro and Cyanidium sp. Sybil Cave, inhabitants of Italian caves [33,34], and the third, Cyanidium sp. Atacama, in a cave on the Chilean coast [35].

The sequences of clones related to the genus Cyanidium identified in Tesoro Cave present identities of 95.09%, 94.75% and 92.76% with the strains from Atacama, Monte Rotaro and Sybil Cave, respectively (Figure 3). In the phylogenetic tree, the Tesoro Cave sequences form a robust clade far from previous Cyanidium isolates suggesting that they could represent a different species of the genus.

Fig 6. Magnification.

Included in the photos.

No. Magnification is not included in the pictures. A scale bar is. But you must include the objective you were using (60x? 100x?) and multiply it by the ocular magnification (10 or 20), so that you must state “Magnification 1000x”.

Please add the magnification used. Write this information in the figure caption.

See previous comments. Scale bar included in figure caption.

L240: most of this section belongs to material and methods.

Moved to Materials and Methods

It is not clear if not in the material and methods section.

L316: most of this section belongs to Material and Methods

These are NOT results. they are MIXED. This information does not belong here. Must be stated in M&M. And the results, in results section.

Moved to Materials and Methods. See lines 150-180.

Reviewer 3 Report

I agree with many of the authors' considerations. Biofilms on cultural heritage stones are very complex and formed by many microbial communities and essential oils, as well as other natural biocides, are specific for certain groups of microorganisms. The use of gels, as matrices to encapsulate biocides, is becoming one of the most important tools for the conservation of cultural heritage, even if the use is limited to laboratory experiments and/or to small surfaces.

As stated by the authors, “restoration companies do not include EOs or natural biocides in their portfolio, due to the difficult or impossible application in the field (large surface extensions, time-consuming protocols, etc.) of previously tested laboratory assays in small stone pieces”.

Oxidizing agents, such as hydrogen peroxide or hypochlorite ions, are known to be low-cost and powerful biocidal agents capable of effectively degrading the organic matter of which microbial species are composed and then this manuscript lacks originality.

I believe that the role of a researcher is to find alternative solutions, for example new eco-friendly broad-spectrum biocides and/or protocols suitable for in situ applications. Only in this way, it will be possible to obtain approval from the competent cultural authorities and restoration companies will be able to use more ecological cleaning procedures. Will it take a long time? Maybe, but I think this is the purpose of our work!

Author Response

Reviewer 3

I agree with many of the authors' considerations. Biofilms on cultural heritage stones are very complex and formed by many microbial communities and essential oils, as well as other natural biocides, are specific for certain groups of microorganisms. The use of gels, as matrices to encapsulate biocides, is becoming one of the most important tools for the conservation of cultural heritage, even if the use is limited to laboratory experiments and/or to small surfaces.

As stated by the authors, “restoration companies do not include EOs or natural biocides in their portfolio, due to the difficult or impossible application in the field (large surface extensions, time-consuming protocols, etc.) of previously tested laboratory assays in small stone pieces”.

Oxidizing agents, such as hydrogen peroxide or hypochlorite ions, are known to be low-cost and powerful biocidal agents capable of effectively degrading the organic matter of which microbial species are composed and then this manuscript lacks originality.

I believe that the role of a researcher is to find alternative solutions, for example new eco-friendly broad-spectrum biocides and/or protocols suitable for in situ applications. Only in this way, it will be possible to obtain approval from the competent cultural authorities and restoration companies will be able to use more ecological cleaning procedures. Will it take a long time? Maybe, but I think this is the purpose of our work!

Thanks for your comments. At present the use of EOs in great surfaces with complex biofilms such as a whole cave has not been sufficiently tested to be included in the restoration company’s portfolio. It is expected that further researches will provide more friendly methods.
